# Urban Vulnerability and Adaptation Strategies against Recurrent Climate Risks in Central Africa: Evidence from N'Djaména City (Chad)

Ndonaye Allarané [1,2,*], Vidjinnagni Vinasse Ametooyona Azagoun [1], Assouhan Jonas Atchadé [1],
Follygan Hetcheli [2] and Joanes Atela [3]

1   Regional Center of Excellence on Sustainable Cities in Africa (CERViDA-DOUNEDON/UL),
    Lomé 01 BP 1515, Togo
2   Research Laboratory on Spaces, Exchanges and Human Security (LaREESH), University of Lomé,
    Lomé 01 BP 1515, Togo
3   African Research and Impact Network (ARIN), Nairobi P.O. Box 53358, Kenya
*   Correspondence: ndonayefils2000@gmail.com

**Abstract:** Climate change and its corollaries suggest that urban planning tools and strategies need to integrate adaptation and resilience approaches into urban development. This study aims to inform decision makers and the scientific community of the importance of appropriating data on urban adaptation and resilience strategies in the city of N'Djaména. After sampling 519 city dwellers, oriented questionnaires and focus groups were used to collect socio-demographic parameters, major climate risks, their impacts on urban issues and the urban resilience strategies employed. The various exposure and impact indices were used to identify and prioritize climate risks and urban exposure issues with the populations concerned. The study highlighted three major climatic hazards, namely, flooding, heat waves and strong winds, and their impact on social and community facilities, the living environment and human health. Ten vulnerability factors have been identified, of which the intrinsically geophysical factors are most familiar to the city's population. The principal component analysis (PCA) illustrates ten (10) strategies for adaptation and resilience to urban climate risks. To meet the climatic challenges in urban areas, this study makes several short-, medium- and long-term recommendations to decision makers.

**Keywords:** climate risks; urban adaptation strategies; vulnerability factors; urban adaptation planning





## 1. Introduction

### 1.1. Urbanization and Climate Change

The report by the Intergovernmental Panel on Climate Change provides irrefutable scientific evidence that climate change will endanger billions of people, affecting livelihoods and essential infrastructure [1]. Cities, in this case in Africa, are undergoing dramatic spatial, social, economic and environmental transformations due to resource availability [2,3], technological innovations [4], global capital flows and climate change [5,6]. Estimated to hold around 60% of the continent's population by 2050 [7], African cities are particularly affected by spontaneous, uncontrolled and environmentally damaging urbanization, making it difficult to manage them sustainably over time and space [8,9]. According to Li et al. [10], rapid urbanization affects populations in different ways, and some of them have become more vulnerable to the effects of climate change. For more than two decades, the international community as a whole has given high priority to environmental issues, with a particular focus on climate change [11], which today represents one of the major challenges for sustainable development [12,13]. As a result of the urbanization process, many urban areas have become vulnerable to climate risks [13,14]. As a result, decision

makers and planners are now being called upon to make risk mitigation and adaptation to climate change essential components of urban planning [2,15].

### 1.2. Vulnerability and Impacts of Climate Change on Urban Issues

According to Zhou et al. [6], vulnerabilities to climate change differ between rural and urban areas due to factors relating to climate change, equipment and professional activities. Urban environments modify the thermal and radiative characteristics of the urban fabric more locally [16]. While cities are partly responsible for climate change through their greenhouse gas-emitting activities, they are also victims of the upsurge in devastating extreme weather events [17]. These have direct effects on a city's physical infrastructure, its buildings, road network, sewage system and energy supply, which in turn impact the well-being and livelihoods of its inhabitants [10]. Furthermore, it has been shown that the consequences for cities will be extreme even if the Paris Agreement target is met, and the temperature rise is kept below 1.5 °C or 2 °C [17]. Vulnerability takes into account both exposure and adaptive capacity [6,18]. Thus, degrees of exposure and sensitivity influence a system's adaptation to climate change [19]. Climate change is adding unexpected challenges to urban areas [20,21], from climate-induced floods, droughts and heat waves [22,23].

Although previous work in Africa has attempted to point out and redirect unsustainable urbanization patterns, the results of numerous scientific studies and the heartfelt cries of many political decision makers emphasize that research efforts must, in particular, focus more on climate change and urban adaptation if development is to be sustainable in the context of climate change, because the current pace of urbanization offers a very time-limited opportunity for cities to adapt [24]. Temperature and rainfall patterns are disrupted by climate change [25], with a considerable scale of extreme weather events.

Increased heat leads to heart disease, particularly in the elderly, pregnant women and people who are already morbidly obese [26]. Van de Walle et al. [27] addressed the aspect of heat stress risks in African cities. Heat-related vulnerability was presented by Jagarnath et al. [28]. Rising temperatures mean high water consumption, which could lead to water-borne diseases such as cholera and respiratory illnesses [29]. Inadequate ventilation in dwellings exposes people to heat-related illnesses [10]. Moda et al. [30] looked at the case of urban workers in their work. A study carried out in Ibadan, Nigeria, showed that human responses to changes in the urban thermal environment with rising temperatures involve several negative aspects [31].

On the other hand, floods and strong winds have a large number of negative impacts on urban social and community facilities, housing and the urban environment [14,32–34]. In view of all the above, Bambara et al. [35] and Wreford et al. [36] consider it necessary for populations to adapt to climate change, as their survival depends on it. Thus, Jamali et al. [32] believe that the question of individual adaptation should be focused on specific studies.

### 1.3. Taking Climate Change into Account in Chad

In Chad, the resurgence of extreme weather phenomena (droughts, floods, heat waves, strong winds, etc.) is one of the highlights of climate change in recent decades [37]. Chad is part of the continent, and more than 1,324,524 inhabitants, or 21.1% of the total population, live in the capital of N'Djaména and are affected by the aforementioned climatic hazards. N'Djaména alone accounts for 40% of the urban population, representing urban growth of around 4.8% [38]. The National Adaptation Action Plan (NAPA) and various national communications point to the many climate risks facing rural and urban populations. At the same time, it is clear that the studies carried out to date on climate change in Chad have been linked to economic development sectors, such as agriculture, livestock, fisheries, health, water and food security. For example, the work of Abdoulay et al. [39] focused on climate variability, perceptions and farmers' adaptation strategies in Chad. The doctoral work of Laohoté Baohoutou [40], for its part, was based on rainfall in the

Sudanian zone of Chad over the last four years, with analysis of the climatic variability of rainfall and mean annual temperature and with sequences of recurrent floods and droughts in the south of the country. Bedoum et al. [41] assessed the vulnerability of agricultural production to rain and drought. Global warming and migration to the shores of Lake Chad were the focus of another study by Kitoto [42]. According to information contained in Chad's Third National Communication on Climate Change [38], the housing sector is already subject to two main climate threats, making it vulnerable to extremely high floods, high temperatures and heat waves, thus exposing habitats and populations. The warning system, the relocation of populations and their habitats and the air-conditioning of offices are the recourse strategies. According to the updated National Determined Contribution [37], the vision of an emerging Chad with a climate-resilient, low-carbon development path focuses on the water, agriculture/agroforestry, livestock and fisheries sectors. It is clear from this page of the National Determined Contribution (NDC) that the urban and infrastructure sectors have been left behind. The strategic axes of the National Policy on Spatial Planning, Urban Development and Housing [43] have not highlighted aspects relating to vulnerability assessment, integration adaptation and urban resilience in their priorities. From the literature review conducted, no scientific study has yet focused on the production of scientific knowledge on climate change in urban environments, in this case in the city of N'Djaména, the vulnerability of its population and the adaptation and resilience strategies used by city dwellers, despite the fact that many scientific works suggest that adaptation and resilience approaches should be taken into account in urban development [6,10,15].

*1.4. Study Rationale*

For the integration of urban adaptation and resilience strategies into urban planning tools and approaches, an important step could be the production of scientific knowledge on cities, their vulnerability as perceived by city dwellers, and current urban adaptation and resilience strategies, albeit of low or medium effectiveness [15]. This will enable decision makers to identify the contours and points of view of city dwellers on climate hazards and their repercussions in the governance and future policies of cities. This is because, in general, better governance and the integration of vulnerability, impacts and adaptation into policies are considered in order to provide facilities that improve people's living conditions [44]. The knowledge and practices that residents hold in relation to how to survive are often not taken into account in cities and countries [45,46], and it is for this reason that the Intergovernmental Panel on Climate Change (IPCC) [1], in its 6th report, chapter 6, stressed the importance of including residents' perspectives [1].

Given this lack of scientific knowledge to support the formulation and implementation of effective urban resilience policies, this study is justified. Thus, its general objective is to fill the gaps in the availability of scientific knowledge on the effects of climate change and the urban adaptation and resilience strategies used in order to activate and catalyze the alignment of climate policies and future urban planning for the sustainability of the city of N'Djaména, those of Africa and even the world. More specifically, this involves the following:

- Using a participatory urban approach, identify urban climate risks and their impact on the challenges facing the city of N'Djaména;
- Index, in a consensual and inclusive manner, the vulnerability factors relating to urban dwellers and urban development units (infrastructure, superstructure and housing, the living environment and human health) in the city of N'Djaména;
- Identify urban adaptation and resilience strategies currently in use to facilitate short-, medium- and long-term recommendations for make cities inclusive, safe, sustainable and resilient.

Achieving these specific goals will help catalyze sustainable urban development and the realization of Sustainable Development Goal 11 of the United Nations Agenda 2030 and the goals of Agenda 2063 relating to the African cities we want [47].

## 2. Materials and Methods

### 2.1. Study Area

N'Djaména, the political capital of Chad, is located between 12°00′ and 13°00′ north latitude and 15°00′ and 16°00′ east longitude. N'Djamena is bordered to the north by the Hadjer-Lamis region, to the east by the Chari-Baguirmi region, to the southeast by the Logone River and to the west by Cameroon (Figure 1). The city is subdivided into 10 municipal districts, with a population of 993,492 in 2009 [48] and an estimated 1,699,208 in 2020 [49].

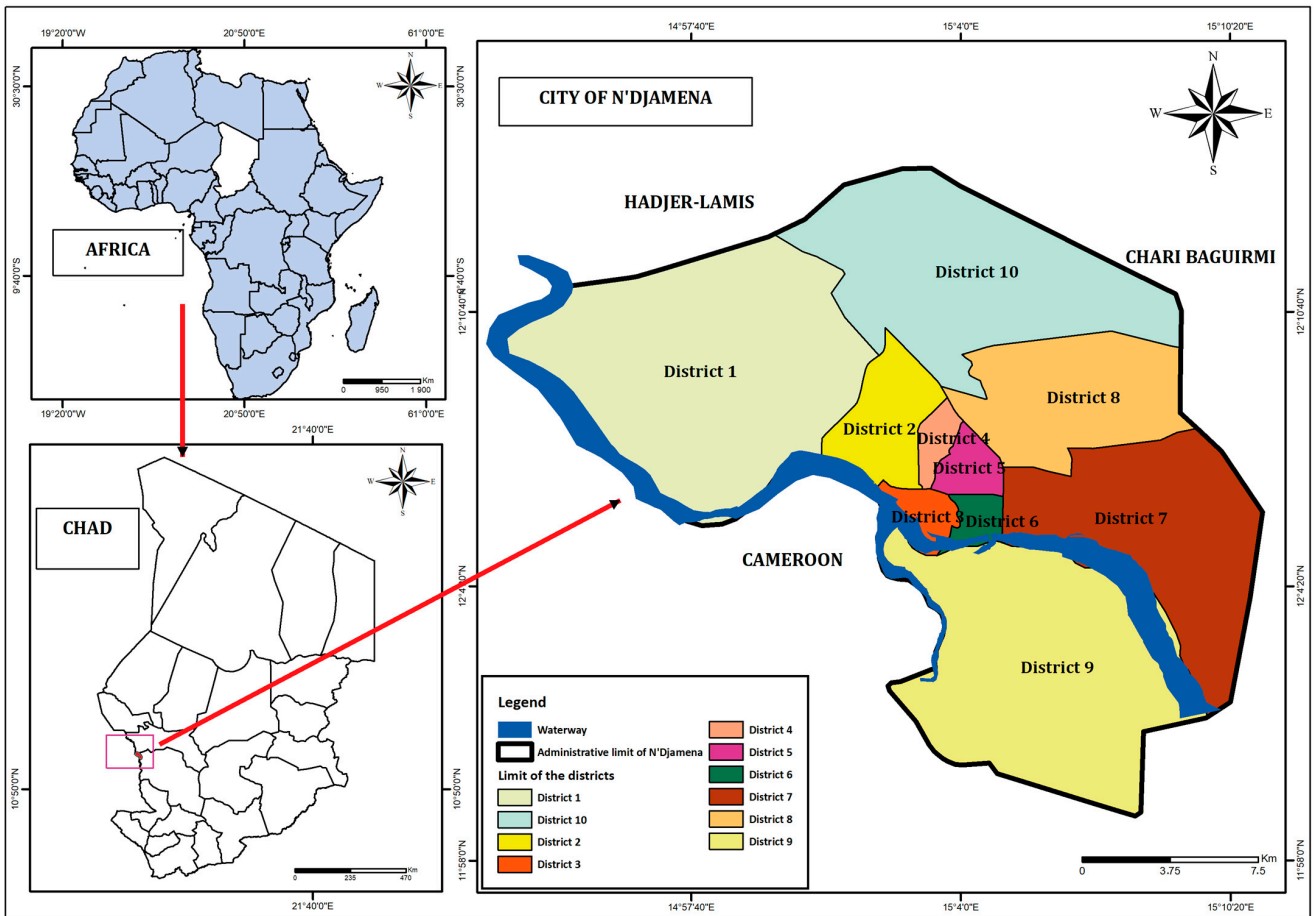

**Figure 1.** Geographical location of the city of N'Djaména.

The city of N'Djaména is characterized by a slightly flat relief, made up of bulges and flood plains. N'Djaména's climate is characterized by six (6) dry months (November to April) and a wet period from May to October. Maximum monthly rainfall (184 mm) is recorded in August (Figure 2). Rainfall does not exceed 700 mm/year and takes the form of more less strong showers [49]. The average monthly temperature ranges from 23.21 °C in January to 33.60 °C in April, with a median of 28.69 °C.

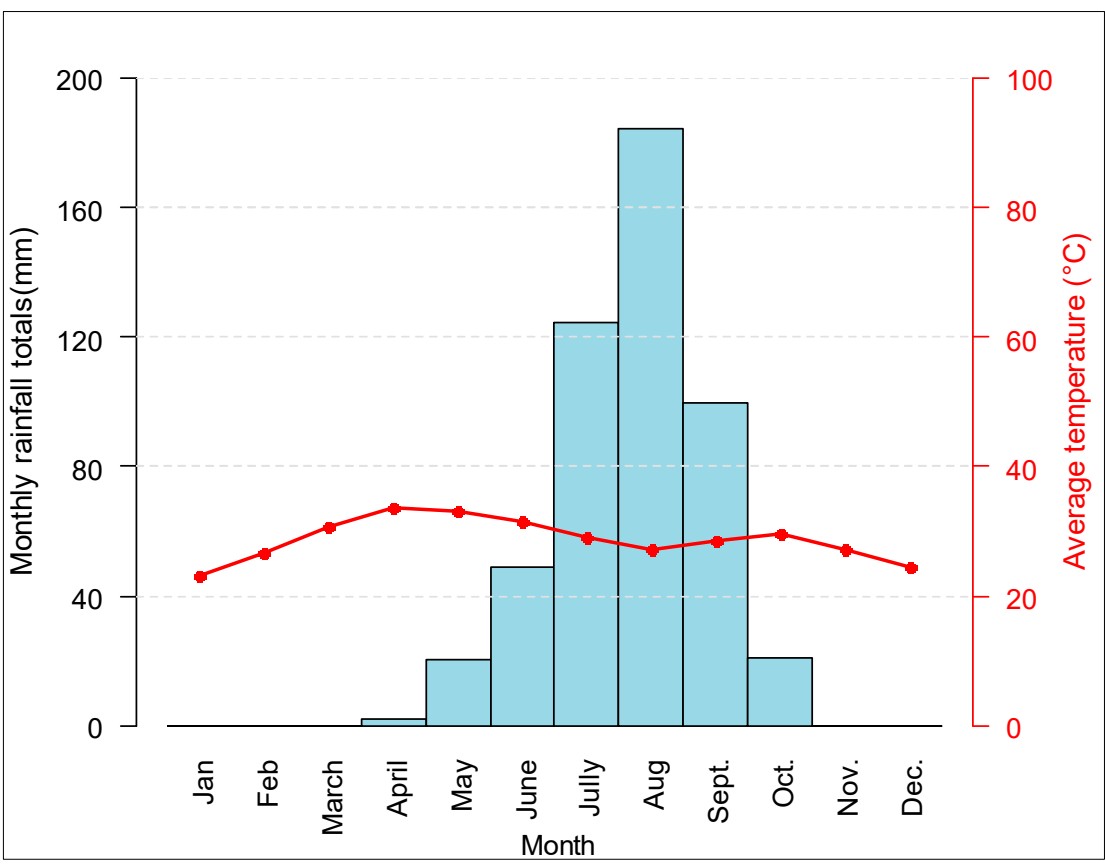

**Figure 2.** Umbrothermal curve for the city of N'Djaména.

*2.2. Sampling and Data Collection*

To carry out this study, we adopted a methodological approach based essentially on literature exploration, individual interviews with the population and focus groups with urban management leadership (Figure 3). The literature review provided us with a panoramic view of the methodologies of similar studies carried out in other countries. Firstly, preliminary investigations with leadership (district managers, municipal decision makers, civil society representatives, etc.) were carried out in each of the city's ten (10) districts. This first stage enabled us to identify and validate the climate risks affecting urban areas. Most of these climate risks can be found in climate policy documents such as the NDC, the NAPA and various national communications from Chad. The work carried out with urban leadership during the focus groups enabled us to list the factors contributing to vulnerability to climate risks, based on the use of the IPCC's Vulnerability, Impacts and Adaptation (VIA) study approach. This approach integrates the climate risk identification and prioritization matrix (through exposure and sensitivity indices), the impact matrix (which highlights the impacts of climate risks and the various urban exposure issues) and, finally, the dimensions of adaptive capacities. To assess the sensitivity of urban issues to climate risks, three decision-making indicators were taken into account: exposure index, impact index and rank. The exposure index (IE) is the sum of the impacts of climate risks on an urban issue. The impact index (II) is the sum of the impacts of a climate risk on urban issues. Rank is used to rank urban issues. The issue with the highest exposure index is the most sensitive to risk.

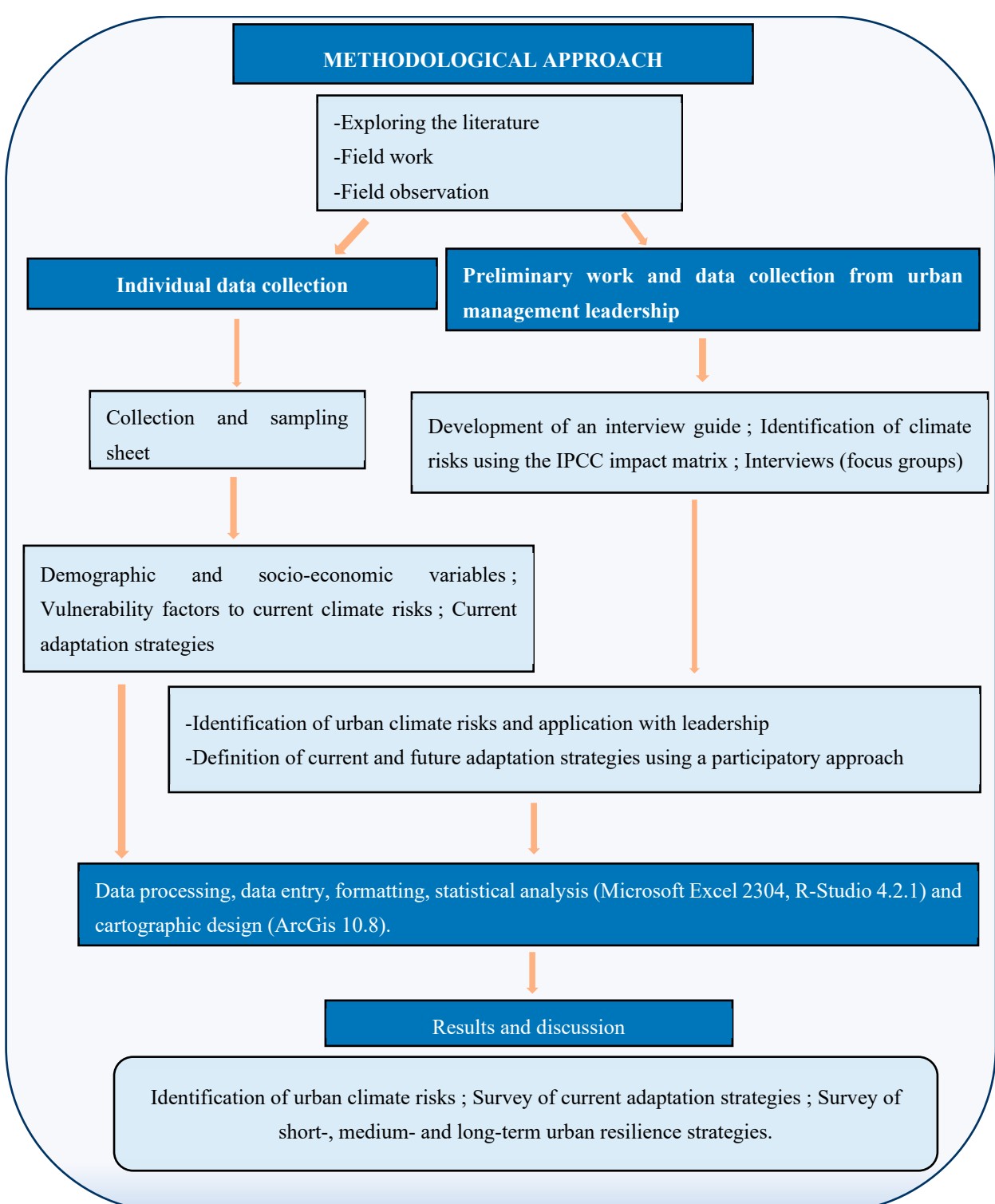

**Figure 3.** Conceptual framework of the study.

This approach was used in the scientific work reviewed by [6] in East and West Africa. It should be noted that the collection sheets used for the individual interviews contained information such as the following:

- Socio-economic characteristics (age, gender, profession, level of education, etc.);
- The various climatic risks identified and validated during the preliminary work, the population's point of view on existing climatic risks;

- Factors contributing to vulnerability to climate risks;
- Impacts and, finally, current urban adaptation strategies used by populations to cope.

A total of ten (10) focus groups, each comprising at least twelve (12) people, were conducted, with one focus group per district. The survey population was spread across the ten (10) districts of N'Djaména. The simple random sampling method was used to select the populations to be interviewed. The sample size was determined using Schwartz's formula, which enabled an inference to be made about the entire urban population of N'Djaména. This formula is illustrated as follows:

$$n_q = \frac{\left[(z_a)^2 \ x \ P(1 - P)\right]}{d^2} \tag{1}$$

With $z_a$: fixed deviation or reduced deviation at a risk of 5% (1.96), corresponding to a confidence interval of 95%; $d$: the margin of error set at 8%; and $P$: the proportion of households per district. Table 1 shows the number of respondents by district.

**Table 1.** Distribution of respondents by district in the city of N'Djaména.

| N'Djaména | Number of Respondents | Percentage (%) |
|---|---|---|
| District 1 | 40 | 8 |
| District 2 | 40 | 8 |
| District 3 | 25 | 5 |
| District 4 | 45 | 9 |
| District 5 | 55 | 10 |
| District 6 | 31 | 6 |
| District 7 | 110 | 21 |
| District 8 | 88 | 17 |
| District 9 | 42 | 8 |
| District 10 | 43 | 8 |
| Total | 519 | 100 |

To facilitate data collection, processing and analysis, the KoboToolbox platform [50] was used and administered using KoboCollect v2022.4.4 software. The interview questionnaire was designed on the basis of the socio-demographic and climatic variables listed above.

*2.3. Data Processing and Analysis*

For data processing and analysis, Microsoft Excel version 2304 was used to arrange and format the data collected from the KoboToolbox platform. Next, R-Studio 4.2.1 software was used with special packages to process the data according to the specific objectives. Khi2 and Fisher's statistical tests were used to achieve certain precision and decision objectives. The presentation and illustration map of the study area was produced using ArcGis software, based on geographic data (in shapefile format) retrieved from the Geographic Information System (GIS) department of N'Djaména town hall.

**3. Results**

*3.1. Climate Risks and Exposure Issues Identified in the City of N'Djaména*

Table 2 presents the results of the identification and prioritization of climate risks and urban facilities exposed to impacts. In the city of N'Djaména, at the end of the focus group work with urban management leadership, the main urban facilities on which populations depend can be summarized as the living environment and human health, superstructures and housing, and infrastructure. Analysis of the sensitivity matrix of these urban issues to the climate risks perceived by the population showed that the living environment and human health (IE = 14) is the first issue exposed to climate risks (Table 2). Infrastructure

(IE = 11) is the least exposed overall. This matrix led to a hierarchy of risks based on the ranking matrix.

**Table 2.** Identification and prioritization of climate risks and urban facilities exposed to impacts in the city of N'Djaména.

| Exposure Units | Climate Risks | | | | |
|---|---|---|---|---|---|
| | Flooding | Heat Waves | Strong Winds | Exposure Index | Rank |
| Infrastructure | 5 | 4 | 2 | 11 | 3 |
| Superstructure and housing | 5 | 3 | 4 | 12 | 2 |
| Living environment and human health | 5 | 5 | 4 | 14 | 1 |
| Impact index | 15 | 12 | 10 | | |

1: No influence; 2: fairly low influence; 3: moderate influence; 4: fairly strong influence; 5: strong influence.

An analysis of Table 3 shows that flooding is the climatic risk with the greatest impact on the livelihoods of the people of N'Djaména. In addition, heat waves and strong winds are also risks.

**Table 3.** Prioritizing climate risks using the impact index.

| Climate Risks | Impact Index | Percentage | Rank |
|---|---|---|---|
| Flooding | 15 | 40.54 | 1 |
| Heat waves | 12 | 32.43 | 2 |
| Strong winds | 10 | 27.03 | 3 |

*3.2. Homogeneous Climate Risk Perception Groups by Frequency*

Table 4 describes the perception of climatic risks in the city of N'Djaména according to the criteria used.

The first group of respondents includes 78.84% educated, 71.25% public sector employees, 63.67% the elderly and 85.56% men. The perceived regular or recurring climatic risks are regular flooding (61.68%) and intense heat waves (74.34%). Similarly, strong winds are statistically average (50.21%, $Pv < 0.001$; v-test > 3). According to the declarations of the city dwellers in this homogeneous group, floods become regular and frequent during rainy periods, while heat waves become regular and intense as the average temperature rises. On the other hand, according to the same group, strong winds are becoming more and more average in the city of N'Djaména (Table 4).

The second group of respondents is made up of 80% young people, 71.25% adults, 78.23% and 70.21% urban high school students and schoolchildren, respectively, 65.55% craftspeople, etc. These respondents have less detailed knowledge of the materialization of the climatic risks cited than the others. According to them, floods and heat waves are the climatic risks significantly ($Pv < 0.001$; v-test > 3) recognized. In this group, 56.85% ($Pv < 0.001$; v-test > 3) of individuals noted the presence of strong winds in the city of N'Djaména.

In summary, this analysis reveals that most of the city dwellers interviewed in N'Djaména have a significant perception of certain climatic risks, the most widely recognized of which are extreme weather events leading to flooding and heat waves. As for their nature, floods are becoming more frequent and regular, as are heat waves. Strong winds are statistically regular only in the perception of the first group of respondents. The parameters that significantly distinguish the perception of different climatic risks are age, gender and occupation. According to N'Djaména residents, the level of literacy also influences people's perception of climate risks.

**Table 4.** Respondent groups for climate risks in N'Djaména.

| Urban Climate Risks and Sociology | Cla/Mod | *p*-Value | v-Test |
|---|---|---|---|
| **1st Respondents Group (RG1)** | | | |
| **Socio-Demographic** | | | |
| Graduation = Superior | 78.84 | <0.001 | 7.25 |
| Occupation = Public employee | 71.25 | <0.001 | 7.64 |
| Age = Older | 63.67 | <0.001 | 3.45 |
| Age = Adult | 28.65 | 0.061 | 1.36 |
| Occupation = Private employee | 80.00 | <0.001 | 4.77 |
| Sex = Male | 85.56 | <0.001 | 6.95 |
| **Floods** | | | |
| Regular | 61.68 | <0.001 | 7.04 |
| Medium | 25.53 | 0.610 | 2.42 |
| Rare | 12.29 | 0.501 | 1.75 |
| **Heat waves** | | | |
| Regular | 74.34 | <0.001 | 8.25 |
| Medium | 24.10 | 0.502 | 2.77 |
| Rare | 1.66 | 0.821 | 1.12 |
| **Strong Winds** | | | |
| Regular | 48.68 | 0.001 | 6.08 |
| Medium | 50.21 | <0.001 | 6.42 |
| Rare | 1.11 | 0.821 | 0.61 |
| **2nd Respondents Group (RG2)** | | | |
| **Socio-Demographic** | | | |
| Age = Young | 80.00 | <0.001 | 8.77 |
| Occupation = Unemployed | 76.72 | <0.001 | 4.64 |
| Graduation = Secondary | 78.23 | <0.001 | 5.51 |
| Graduation = Primary | 70.21 | <0.001 | 4.35 |
| Age = Adult | 71.25 | <0.001 | 4.38 |
| Occupation = Craftspeople | 65.55 | <0.001 | 5.26 |
| **Floods** | | | |
| Regular | 20.39 | 0.601 | 3.62 |
| Medium | 77.06 | <0.001 | 8.04 |
| Rare | 2.65 | 8.652 | 1.27 |
| **Heat waves** | | | |
| Regular | 30.30 | 0.503 | 2.97 |
| Medium | 58.91 | <0.001 | 3.67 |
| Rare | 10.90 | 0.836 | 1.01 |
| **Strong winds** | | | |
| Regular | 16.13 | 0.513 | 2.48 |
| Medium | 32.00 | 0.001 | 2.88 |
| Rare | 56.85 | 0.015 | 2.46 |

Cla/Mod: percentage of respondents; *p*-value: significance level of analysis; v-test: measures association between variables and groups. When the v-test is >3, the response is significant.

### 3.3. Influence of Social Parameters on the Perception of Vulnerability to Climate Risks in the City

Figure 4 shows the cross-tabulation of social parameters and vulnerability factors. Analysis of this figure shows that the proportion of the vulnerability factor "VFGP" is higher than that of the other factors, whatever the social category to which it belongs. The values of the standardized residuals of the proportions of these cross-tabulations are between −2 and 2, i.e., very close to the numbers expected under the independence hypothesis, with the exception of adults (b) and retailers (b), who appear to be represented with residual values greater than or equal to 2.4 (blue VFGP boxes).

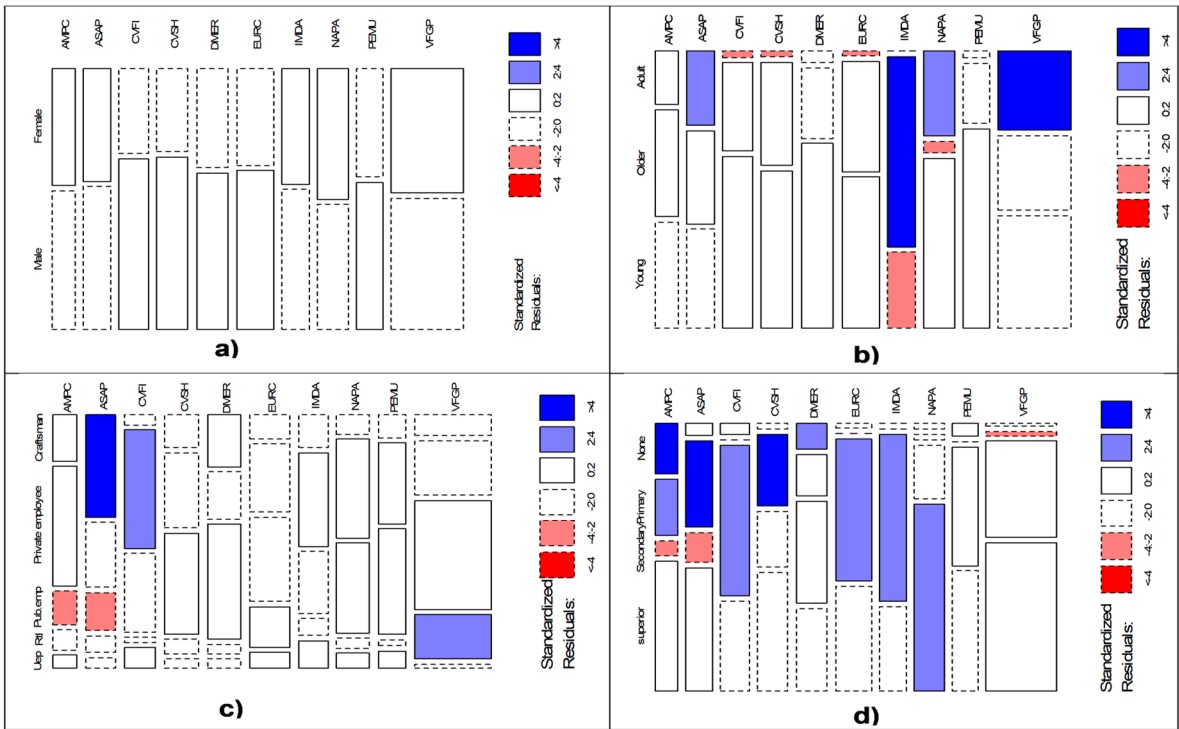

**Figure 4.** Graphic representation of the intersection of social parameters and vulnerability factors. Vulnerability factors identified: lack of an early warning system for climatic risks specific to the city (ASAP); lack of disaster recovery mechanisms (AMPC); inadequate mechanisms and provision of emergency humanitarian assistance (IMDA); deficiencies in economic response mechanisms to urban climate disasters (DMER); economic precarity of urban households (PEMU); outdated and fragile infrastructure equipment (CVFI); outdated and fragile superstructure and housing facilities (CVSH); exposure of urban facilities to climate risks (EURC); no adaptation of urban planning policies (NAPA); vulnerabilities due to geophysical factors (VFGP). (**a**) Cross-referencing gender with vulnerability factors. (**b**) Cross-referencing age with vulnerability factors. (**c**) Cross-referencing professional occupations with vulnerability factors. (**d**) Cross-referencing education with vulnerability factors.

The Fisher test applied to the intersection of gender and vulnerability factor shows a *p*-value above the 5% threshold (0.58). This explains why the perception of the factors is in no way linked to gender. This result was confirmed by the residual chi-square test, which indicates that the independence of the vulnerability factors is not linked to probable sample bias. On the other hand, perception of the factors depends on age (*p*-value = 0.0004), grade (*p*-value = 0.0005), profession (*p*-value = 0.0005) and level of education (*p*-value = 0.0005), according to the same Fisher test applied to cross-tabulations.

### 3.4. Impact of Climate Risks on Urban Issues in the City of N'Djaména

Table 5 sets out the impacts of climate hazards on the city's urban challenges, using an impact matrix that highlights the exposure of the city's urban challenges to urban climate hazards. From an analysis of this table, it can be seen that equipment (infrastructure, superstructure and housing), the living environment and human health are affected in different ways, mainly by major meteorological extremes such as flooding, heat waves and strong winds.

**Table 5.** Summary of climate risk impacts from interviews with urban managers.

| Impact Matrix | Urban Resources Impacted | | |
| --- | --- | --- | --- |
| | Infrastructure | Superstructure and Housing | Living Environment and Human Health |
| Flooding | - Road damage;<br>- Damage to water drainage systems;<br>- Impassable roads and drainage systems;<br>- Regular accidents;<br>- Decline in socio-economic activities;<br>- Collapsing network support. | - -Deterioration of collective social facilities;<br>- Reduced and difficult access to collective social services;<br>- -Declining supply of social and professional services;<br>- -Increasing social difficulties and poverty;<br>- - Weakening of the social fabric and generating of socio-political tensions or conflicts, etc. | - Water saturation and stagnation;<br>- Access difficulties and loss of material goods;<br>- Pollution and multiplication vectors of endemic diseases (malaria, cholera, infections) and waterborne diseases;<br>- Increased morbidity and mortality among the vulnerable (the elderly, children, pregnant women, etc.). |
| Heat waves | - Cracking of asphalt roads;<br>- Road accidents;<br>- Acceleration of polluting chemical reactions in drainage systems;<br>- Increase in health problems. | - Warming work areas and offices;<br>- High demand for energy consumption for ventilation and air conditioning;<br>- Decline in household and national economies;<br>- Increased health risks. | - Frequent rise in temperature;<br>- Reduced professional and social productivity;<br>- Decline in municipal and national economies;<br>- Physiological dehydration;<br>- Risk of cardiovascular incidents;<br>- Insomnia in the home. |
| Strong winds | - Collapsing network support;<br>- Reduced community services;<br>- Fragilization of the social fabric and generation of socio-political tensions and conflicts. | - Assignment, deterioration and disappearance of community facilities;<br>- Reduced community services;<br>- Increased risk of accidents and fatalities;<br>- Generating socio-political tension and conflict. | - Pollution of the living environment;<br>- Distribution of germs and pollutants (aerosols, heavy metals, pollens, dust);<br>- Contamination of mucous membranes and biological cavities;<br>- Recurrence of lung disease;<br>- Community health assignments;<br>- Increased morbidity and mortality. |

- Impact of flooding on urban service facilities

According to the statements made by the management elite and the people of N'Djaména, climatic hazards have many impacts on the city's facilities. Among the impacts identified and mentioned in Table 5, we can note, under the influence of flooding, the deterioration of roads, the deterioration of water drainage networks and the deterioration of collective social facilities. These impacts then generate effects such as the impassability of roads and networks, regular accidents on roads and highways, reduced and more difficult access to social services, a decline in the supply of social and professional services, a drop in socio-economic activities, an increase in social difficulties and poverty, the weakening of the social fabric and the generating of socio-political tensions or conflicts.

- Impact of flooding on health and the urban environment

According to the statements made by the management elite and the people of N'Djaména, floods have had a major negative impact on the health of the peaceful population and the city's environment (Table 5). Among the impacts identified and mentioned are water saturation and stagnation in living areas, difficulties of access and loss of material goods, pollution and multiplication of vectors of endemic diseases (malaria, cholera, infections) and waterborne diseases, and increased morbidity and mortality among vulnerable people (the elderly, children, pregnant women, etc.).

- Impacts of heat waves and strong winds on health and the urban environment

Heat waves have varying degrees of impact on human health and the living environment. According to the results of the matrix of climate risk impacts on urban issues (Table 5), heat waves cause an increase in the frequency of average temperatures, a drop in professional and social productivity, a decline in the municipal and national economy, physiological dehydration, the risk of cardiovascular accidents and insomnia in the home. As for strong winds, they cause pollution of the living environment, the distribution of germs and pollutants (aerosols, heavy metals, pollens, dust) in the surrounding environment, contamination of mucous membranes and biological cavities, and recurrence of lung disease and death.

*3.5. Adaptation and Resilience Strategies for Major Climate Risks in the City*

- Correlation between adaptation strategies and major climate risks

Table 6 presents the results of the correlation of major climate risks and urban adaptation and resilience strategies with the first two principal dimensions of the principal component analysis (PCA). On the one hand, two urban climate risks are negatively and positively correlated with the first dimension (Dim1) of the PCA. These are, respectively, floods and heat waves, known as the extreme weather phenomena that impact urban populations and challenges (Table 6). Similarly, two homogeneous pools of adaptation strategies and urban resilience are negatively and positively correlated with this first PCA dimension. The first homogeneous pool, made up of the DDT, UME, UPI and CDU strategies, is negatively associated with the first dimension, while the second pool, which includes strategies such as DHM, UEV, PRE and USE, is positively linked to the same dimension (Table 6). On the other hand, strong winds, which are also perceived as extreme weather phenomena, are positively correlated with the second dimension of the PCA. Similarly, the RET and ASI adaptation and resilience strategies are both associated with this second dimension (Table 6).

**Table 6.** Correlation of major climate hazards and urban adaptation and resilience strategies with the first two main PCA dimensions.

| Climate Risks and Adaptation Strategies | Dim1 | Dim2 |
| --- | --- | --- |
| Flooding | −0.68 | 0.28 |
| Strong winds | 0.03 | 0.91 |
| Heat waves | 0.73 | 0.30 |
| DDT | −0.71 | 0.28 |
| ASI | −0.51 | 0.54 |
| UME | −0.59 | 0.32 |
| UPI | −0.55 | 0.48 |
| PRE | 0.25 | 0.74 |
| CDU | −0.52 | 0.37 |
| RET | −0.25 | 0.92 |
| USE | 0.12 | 0.66 |
| DHM | 0.24 | 0.53 |
| UEV | 0.35 | 0.62 |

RET: Refurbishing equipment after strong windstorms; ASI: intercommunity social assistance; USE: use of ecosystem services from trees (e.g., tree shade) on hot days; DHM: sleep outside the house (in the open air) on hot nights; UEV: energy use for ventilation of homes and offices against heat waves; PRE: regular intake of drinking water; DDT: temporary relocation of households after flooding; UME: use of motor-driven pumps to evacuate water after flooding; UPI: use of traditional pirogues during floods; CDU: building of urban dikes.

- Correlation between climate risks and urban adaptation and resilience strategies used

    Figure 5 shows the results of the principal component analysis (PCA) plane projection. The projection of the three (3) major climatic risks (floods, strong winds and heat waves) and the ten (10) urban adaptation and resilience strategies in the PCA plan indicates that during floods, the populations of the city of N'Djaména resort to adaptation and resilience strategies such as temporary household displacement and relocation (DDT), use of motor-driven water pumps (UME), use of artisanal pirogues during floods (UPI) and the building of urban dikes (CDU) (Figure 5). On the other hand, the same figure implies that using ecosystem services from trees (USE) on hot days, sleeping outside the dwelling (in the open air) on hot nights (DHM), using electrical energy for ventilation (UEV) of dwellings and offices against heat waves are the adaptation and resilience strategies used by city dwellers. As for adaptation and resilience in the face of strong winds, repairing equipment after strong windstorms and installing windbreaks (RET) around homes are the most popular. Intercommunity social assistance (ASI) is a strategy used against both the climatic hazards of flooding and strong winds (Figure 5).

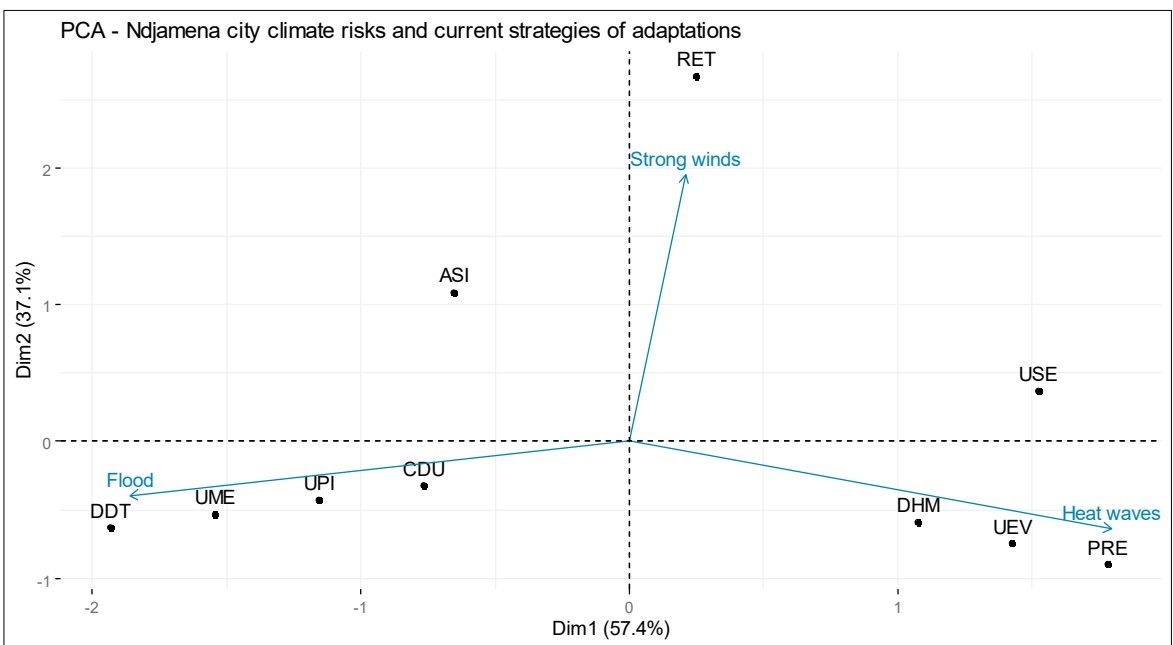

**Figure 5.** Projection in the first two main axes of the PCA of climate risks and the urban adaptation and resilience strategies used. RET: Refurbishing equipment after strong windstorms; ASI: inter-community social assistance; USE: use of ecosystem services from trees (e.g., tree shade) on hot days; DHM: sleep outside the house (in the open air) on hot nights; UEV: energy use for ventilation of homes and offices against heat waves; PRE: regular intake of drinking water; DDT: temporary relocation of households after flooding; UME: use of motor-driven pumps to evacuate water after flooding; UPI: use of traditional pirogues during the floods; CDU: building of urban dikes.

## 4. Discussion

### 4.1. Socio-Demographic Variables, Climate Risks and Vulnerability to Climatic Hazards

- Socio-demographic variables, climatic risks

The parameters that significantly distinguish the perception of different major climatic risks (floods, heat waves and strong winds) are age, gender and occupation. According to N'Djaména residents, literacy level had no influence on their perception of climate risks. According to Michel-Guillou [51], the theory of social representations is particularly well suited to understanding how individuals represent, position and act towards environmental problems.

- Socio-demographic variables and vulnerability to climatic hazards

The perception of vulnerability factors depended on age ($Pv$ = 0.0004), professional occupation ($Pv$ = 0.0005) and level of education ($Pv$ = 0.0005), according to the same Fisher test applied to cross-tabulations. These results are in line with the work of Zhou et al. [6], Michel-Guillou [51] and Williams et al. [52]. The work of Hlahla and Hill [53] highlighted the fact that marginalized communities are highly sensitive to climatic hazards.

### 4.2. Impact of Urban Climate Risks on Urban Issues

According to information from the matrix of climate risk impacts on urban issues in the city of N'Djaména, facilities (infrastructure, superstructure and housing), the living environment and human health are affected in different ways by the effects of major meteorological extremes such as floods, heat waves and strong winds. Influences of flooding include deterioration of roads, water drainage networks and social amenities. In the literature review produced by Monteiro et al. [15], it is highlighted that the capacity of urban infrastructure to withstand flooding is very fragile. According to Li et al. [10], devastating extreme weather events have a direct impact on a city's physical infrastructure—its build-

ings, road network, sewage system and energy supply—which in turn has an impact on the well-being and livelihood of its inhabitants. It is therefore necessary, at an engineering level and for future projects, to add a safety factor in this sense to the dimensioning of infrastructure, superstructure and housing. In the chain of impacts, these effects include the impassability of roads and networks, regular accidents on roads and highways, reduced access to social services, reduced supply of social and professional services, reduced socio-economic activities, increased social hardship and poverty, weakening of the social fabric and the generating of socio-political tensions and conflicts. According to Mitchell et al. [20] and Serdeczny et al. [21], flooding is adding unexpected challenges to urban areas and their infrastructure as a result of climate change. As a result, public authorities are being called upon to meet their responsibility to protect populations and their assets in the face of these relentless challenges caused by this global crisis [35,36].

With regard to the impact of flooding on living conditions and human health, the report revealed saturation and stagnation of water in living environments, difficulties of access and losses of material goods, pollution and multiplication of vectors of endemic diseases (malaria, cholera, infections) and waterborne diseases, and increased morbidity and mortality among vulnerable people (the elderly, children, pregnant women). These results support and confirm the realities observed in Chad's urban territory, since according to the information contained in the third national communication [38], the housing and urban planning sector is under the sway of two main climatic threats, flooding and extremely high temperatures and heat waves, exposing habitats and populations alike. What is more, floods and strong winds have major and numerous negative impacts on collective urban social facilities and the urban living environment [14,32].

The study also revealed that heat waves cause an increase in the frequency of average temperature, a decrease in professional and social productivity, a decline in the municipal and national economy, physiological dehydration, the risk of cardiovascular incidents and insomnia in the home. According to Orimoloye et al. [26], increased heat leads to heart disease, particularly in the elderly, pregnant women and people who are already morbidly obese. Van de Walle et al. [27] focus on rising air temperatures in cities, indicating the heterogeneity of vulnerabilities and risks associated with heat stress. Steadily rising temperatures in the city affect the health of residents [29,31]. Moda et al. [30] pointed out the forms of pollution and vector-borne diseases to which workers are exposed.

*4.3. Current Population Adaptation Strategies*

The major climatic risks (floods, strong winds and heat waves) and the ten (10) adaptation strategies projected in the PCA plan indicate that, during floods, the populations of the city of N'Djaména resort to adaptation and resilience strategies such as temporary household displacement and relocation (DDT), use of motor-driven water pumps (UME), use of artisanal pirogues during floods (UPI) and the building of urban dikes (CDU). According to Chad's updated National Determined Contribution [37], populations and public authorities resort to temporary or permanent means of displacement in the event of flooding. These results support the adaptation strategies revealed in the updated NDC. Jamali et al. [32] encourage the development of study programs to replace individual strategies.

Use of ecosystem services from trees (USE) on hot days, sleeping outside the dwelling (in the open air) on hot nights (DHM), the use of electrical energy for ventilation (UEV) of dwellings and work offices are the adaptation and resilience strategies against heat waves used by city dwellers. As for adaptation and resilience in the face of strong winds, repairing equipment after strong windstorms and installing windbreaks (RET) around homes are the most popular. Intercommunity social assistance (ASI) is a strategy used against both the climatic hazards of flooding and strong winds. The work of Atchadé et al. [9] supported the fact that the populations of the city of Cotonou use the urban ecosystem services of trees to defend themselves against climatic hazards such as heat waves and strong winds. The authors showed that the shade and wind-breaking services that urban trees offer city dwellers enable them to adapt. Li et al. [10] have also shown that ventilating

rooms and staying or sleeping outdoors in the evening are strategies that some urban dwellers in African cities use in hot conditions. Using water to cool down and bathe are also alternatives offered by citizens in hot weather [29].

## 5. Conclusions

This study, carried out in the city of N'Djaména, aims to inform decision makers and the scientific community of the importance of integrating climate risk perception, vulnerability factors, impacts and adaptation and resilience strategies developed by citizens in Sahelian cities. The study identifies three major climate hazards (flooding, heat waves and strong winds) which impact urban issues (infrastructure, superstructure, housing, living environment and human health) in various ways. Ten vulnerability factors have been identified, of which the intrinsically geophysical factors are most familiar to the city's populations. The results of the PCA plane projection illustrate ten (10) adaptation and resilience strategies specifically applied in the face of urban climate risks. Each risk is matched by an array of response strategies currently used by residents. To meet urban climate challenges, planners and managers need to focus on the resilience and sustainability of urban facilities, the living environment and human health, in addition to structural changes and new guidelines for municipal climate governance.

*Recommendations*

For the city of N'Djaména, the resilience path proposed in this study is a set of possible resilience solutions drawn from vulnerability factors. These solutions are translated into objectives for reducing vulnerability and increasing resilience in the short, medium and long term.

- Short term

Improving Chad's national early warning system for the above-mentioned climate risks, setting up a community-based early warning system specific to the city of N'Djaména, improving urban care mechanisms, strengthening emergency humanitarian assistance mechanisms and provisions, increasing economic mechanisms for responding to climatic risks and disasters, strengthening economic recovery mechanisms for the urban sector, strengthening the immune system of vulnerable people (the elderly, children and pregnant women) in the city, strengthening urban governance policy, rehabilitating failing infrastructure and superstructure facilities, etc.

- Medium term

Revitalizing disaster risk management structures at the urban (city) level, and better integrating climate and disaster risks into urban planning documents. Strengthening policies for the management of the living environment, urban development, water management and sanitation. Investing in scientific research to train specialists in urban management and natural disasters. Investing in the construction of equipment (infrastructure and superstructure) and incorporating high-albedo materials; strengthening urban and institutional governance, etc.

- Long term

Reinforcing the promotion of a savings culture, promoting and encouraging urban mutual insurance companies, intensifying functional literacy at the outskirts of N'Djaména, creating and strengthening early warning systems for flood risks, creating an interface between meteorological services and the urban population, strengthening and diversifying household sources of income. Improving technologies for the installation and construction of infrastructure and superstructure equipment, as well as housing, etc.

**Author Contributions:** Conceptualization, N.A.; methodology, N.A. and A.J.A.; software, N.A. and V.V.A.A.; validation, F.H.; formal analysis, N.A. and A.J.A.; investigation, N.A.; resources, N.A. and F.H.; data curation, N.A.; writing—original draft preparation, N.A. and A.J.A.; writing—review and editing, N.A.; visualization, F.H. and J.A.; supervision, F.H. and J.A.; project administration, F.H.; funding acquisition, N.A. All authors have read and agreed to the published version of the manuscript.

**Funding:** This research was funded by the Regional Center of Excellence on Sustainable Cities in Africa (CERViDA-DOUNEDON), Association of African Universities (AUA) and the World Bank.

**Data Availability Statement:** Data will be made available on request.

**Acknowledgments:** The authors would like to thank the Regional Center of Excellence on Sustainable Cities in Africa (CERViDA-DOUNEDON), the Association of African Universities and the World Bank Group for the financial support that made this study possible.

**Conflicts of Interest:** The authors declare no conflict of interest.

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
