# Peer review of "Urban Vulnerability and Adaptation Strategies against Recurrent Climate Risks in Central Africa: Evidence from N’Djaména City (Chad)"

_urbansci, doi:10.3390/urbansci7030097_

Round 1

Reviewer 1 Report

This study sought to determine the major impacts of climate change on urban life in the capital city of Chad.  The study found 3 major impacts on urban life, heat, flood, wind and identified the most common resilience strategies undertaken by the population when faced with these problems. 

There are two poorly identified concepts right in the abstract.  1) The article says that “10 vulnerability factors have been indexed…”  I do not know what indexed means in the sentence and it is not clear who is doing the “indexing”, whatever that is.  2) “CPA” is an unexplained abbreviation in the abstract.  Please spell it out.

The text of the piece makes the same error over and over by putting the number of a reference into a sentence as though the meaning of that is clear.  Numbers of references are not used in that way.  In every case, the reference number should go at the end and if the author wishes to say where it came from, that must be part of the sentence.  Otherwise, the fact can be simply stated and the reference number put at the end of the sentence.   The first example of this is in the first paragraph of the paper where it says, “According to [10], rapid urbanization affects populations in different ways, some of which have become more vulnerable to the effects of climate change.” Instead of this, the author could say, “According to Smith and Sharon (use names or refer to them in some other way), rapid urbanization…..effects of climate change. [10]  This error is made repeatedly throughout the paper and must be changed.

Section 1.2 of the paper should start a fresh paragraph at “Although previous work in Africa……”  Also this section would seem to imply a justification for localized analysis of urban climate risks.  But the paragraph doesn’t make that clear.  Isn’t that the reason for focusing in on this city and learning exactly what adaptations are needed?

Another unexplained abbreviation:  “CDN”.

Section 2.2 There is another unexplained abbreviation: PNA.

I am not familiar enough with the Schwartz formula to judge the use in this paper. 

Figure 3 on the Conceptual framework of the study seems flawed.  First box on the right under Field Work, is missing words.  Same with the third box in that series (word missing after participatory).  Didn’t the interviews help to determine the survey?   If not, what is the connection between these two pieces of the project?  The figure suggests that the surveys are not used for the purpose of results and discussion.  If that is true, something is amiss in this methodology.   The figure should be redrawn to reflect the entire project.

3.1 Use of certain terms is confusing.     

               Use of the term “issues” in the sentence “…..urban issues exposed to impacts” is unclear.  It is an odd use of the word “issues”.  Does it mean “threats?” “vulnerabilities?”, “aspects of urban life?”  The next sentence suggests that it means aspects of urban life. 

3.2, 3.3, 3.4  these sections seem to me to be the most important in the paper

3.5   The authors should spell out PCA, as well as the strategies DDT, UME, etc.  Otherwise the meaning of this section is pretty much lost to many readers, including this one.  Alternatively, there could be an instruction like (see figure below for meaning of abbreviations).    

Conclusion:  I am not familiar with any use of the word “bacterium” for this meaning.  This will be lost to American audiences.  Another term should be used here.  Also, again the word “indexed” is an unconventional use.    What is this sentence trying to say, “Ten vulnerability factors have been indexed….” Discussed?  Explored?  Studied? Described?

The paper is longer than it needs to be.  Some of the background could be much briefer, especially where points are made that reflect a current consensus.    

Please see the notes copied and pasted above. 

Author Response

Dear Reviewer

Thank you for taking the time to provide quality comments on our manuscript despite your busy schedule. Taking into account your comments, please find the answers point by point in the lines below.

Point by pont responses

Point 1 : 1) The article says that “10 vulnerability factors have been indexed…” I do not know what indexed means in the sentence and it is not clear who is doing the “indexing”, whatever that is.

Response 1 : We have replaced indexed with identified as you recommend. The sentence becomes : 10 vulnerability factors have been identified. Done

Point 2 : 2) “CPA” is an unexplained abbreviation in the abstract. Please spell it out.

Response 2 : This is called Principal Component Analysis (PCA). We have corrected this. Done

Point 3 : The text of the piece makes the same error over and over by putting the number of a reference into a sentence as though the meaning of that is clear. Numbers of references are not used in that way. In every case, the reference number should go at the end and if the author wishes to say where it came from, that must be part of the sentence. Otherwise, the fact can be simply stated and the reference number put at the end of the sentence. The first example of this is in the first paragraph of the paper where it says, “According to [10], rapid urbanization affects populations in different ways, some of which have become more vulnerable to the effects of climate change.” Instead of this, the author could say, “According to Smith and Sharon (use names or refer to them in some other way), rapid urbanization...effects of climate change. [10] This error is made repeatedly throughout the paper and must be changed.

Response 3 : We have corrected this reference error throughout the document. Done

Point 4 : Section 1.2 of the paper should start a fresh paragraph at “Although previous work in Africa……” Also this section would seem to imply a justification for localized analysis of urban climate risks. But the paragraph doesn’t make that clear. Isn’t that the reason for focusing in on this city and learning exactly what adaptations are needed ?

Response 4 : Done. Indeed, you're right about that pertinent comment.

Point 5 : Another unexplained abbreviation : “CDN”.

Response 5 : The National Determined Contribution (NDC). Done

Point 6 : Section 2.2 There is another unexplained abbreviation : PNA.

Response 6 : You're right. This is the National Adaptation Action Plan (NAPA). The definition is given for the first time in line 104. Done

Point 7 : Figure 3 on the Conceptual framework of the study seems flawed. First box on the right under Field Work, is missing words. Same with the third box in that series (word missing after participatory). Didn’t the interviews help to determine the survey ? If not, what is the connection between these two pieces of the project ? The figure suggests that the surveys are not used for the purpose of results and discussion. If that is true, something is amiss in this methodology. The figure should be redrawn to reflect the entire project.

Response 7 : Thank you for your comments on figure 3 concerning the conceptual framework. It was the layout effect that caused some words to be hidden. We have corrected this. The interviews helped determine the survey. The surveys are of course used for results and discussion purposes. We have also taken into account your recommendations regarding the shape of the figure. Done

Point 8 : 3.1 Use of certain terms is confusing.

Use of the term “issues” in the sentence “….urban issues exposed to impacts” is unclear. It is an odd use of the word “issues”. Does it mean “threats ?” “vulnerabilities ?”, “aspects of urban life ?” The next sentence suggests that it means aspects of urban life.

Response 8 : Done. We have corrected this (…urban facilities)

Point 9 : 3.5 The authors should spell out PCA, as well as the strategies DDT, UME, etc. Otherwise the meaning of this section is pretty much lost to many readers, including this one. Alternatively, there could be an instruction like (see figure below for meaning of abbreviations).

Response 9 : We have inserted the meaning of the abbreviations below the table for a better understanding of the readers. Done

Point 10 : Conclusion : I am not familiar with the use of the word “bacterium” for this meaning. This will be lost to American audiences. Another term should be used here. Also, again the word “indexed” is an unconventional use. What is this sentence trying to say, “Ten vulnerability factors have been indexed…” Discussed ? Explored ? Studied ? Described ?

Response 10 : We have corrected the conclusion

We have replaced indexed with identified as you recommend. The sentence becomes : 10 vulnerability factors have been identified. Done

Best regards.

Reviewer 2 Report

1. why did you bias your sample to the elite?

2. why is there a link with 

3. Your first priority 'living environment'sounds vary vague!

4, what are the superstructures in table 2?

5. what is the link with urban management in the differents stages?

6. Did you use Likert scales?

an interesting paper, bu one senses se translated French

Author Response

Dear Reviewer

Thank you for taking the time to provide quality comments on our manuscript despite your busy schedule. Taking into account your comments, please find the answers point by point in the lines below.

Point by pont responses

Point 1 : Why did you bias your sample to the elite ?

Response 1 : We worked upstream with the urban management elite to identify the main climatic risks. After this preliminary work, we turned to the population for the surveys using survey forms. These surveys are a continuation of the work begun with this elite group

Point 2 : Why is there a link with

Response 2 : This is because they are the people in charge of urban structures (municipal district managers, neighborhood chiefs, urban planning managers, etc.). They are best placed to understand and intervene upstream on issues of climate risk and urban development.

Point 3 : Your first priority « living environment » sounds vary vague

Response 3 : Cadre de vie refers to the urban environment of the city of N'Djaména, i.e. the urban environment in which people live.

Point 4 : What are the superstructures in table 2 ?

Response 4 : The superstructures in Table 2 include social and community facilities such as schools, hospitals, public buildings, etc. These facilities are in the public interest

Point 5 : What is the link with urban management in the differents stages ?

Response 5 : This link with urban management at different stages because today, with climate change issues, these urban managers have to take into account climate risk aspects in the process of drawing up urban planning documents.

Point 6 : Did you use Likert scales ?

Response 6: Yes. For example, Table 2: 1: no influence; 2: fairly low influence; 3: moderate influence; 4: fairly strong influence; 5: strong influence.

Best regards.

Reviewer 3 Report

This is a well written paper that identifies climate change risk factors for N’Djaména city (Chad). Below see some minor suggestions 

(1) line 20, change "such as" to "namely"

(2) Line 91-92, the references below are also good examples for wind and rain effects on urban climate change risks. 

Qin, H., & Stewart, M. G. (2020). Risk-based cost-benefit analysis of climate adaptation measures for Australian contemporary houses under extreme winds. Journal of Infrastructure Preservation and Resilience1(1), 1-19.

Qin, H., & Stewart, M. G. (2022). Adaptation of housing to climate change and extreme windstorms. Engineering for Extremes: Decision-Making in an Uncertain World, 119-141.

(3) The study focuses residents' responses. How important is it for city planners and decison makers?

English is good.

Author Response

Cher critique

Merci d'avoir pris le temps de fournir des commentaires de qualité sur notre manuscrit malgré votre emploi du temps chargé. En tenant compte de vos commentaires, veuillez trouver les réponses point par point dans les lignes ci-dessous.

Réponses point par point

Point 1 : (1) ligne 20, remplacer ''tel que'' par ''à savoir''

Réponses 1 : Terminé. Nous avons remplacé

Point 2  : (2) Ligne 91-92, les références ci-dessous sont également de bons exemples des effets du vent et de la pluie sur les risques de changement climatique urbain.

Qin, H., & Stewart, MG (2020). Analyse coûts-avantages basée sur les risques des mesures d'adaptation au climat pour les maisons contemporaines australiennes soumises à des vents extrêmes. Journal of Infrastructure Preservation and Resilience, 1  (1), 1-19.

Qin, H., & Stewart, M.G (2022). Adaptation of housing to climate change and extreme windstorms. Engineering for Extremes : Decision-Making in an Uncertain World, 119-141.

Response 2 : Thank you for your two references. The content of these two documents is very rich in information. We have also quoted from your recommendations in the document. We'll be using them again in our next article, which will focus exclusively on the effects of wind and rain in urban environments.

Point 3 : (3) The study focuses resident’s responses. How important is for city planners and decison makers ?

Réponse 3 : L'étude porte non seulement sur les réponses des habitants, mais aussi sur les réponses de l'élite de la gestion urbaine (cette élite comprend les décideurs municipaux, les chefs de quartier, les représentants de la société civile, etc.) qui interviennent principalement sur les questions d'intérêt à la population. Cette étude est importante pour les urbanistes et les décideurs, car elle leur permet désormais de prendre en compte les enjeux des risques climatiques dans l'élaboration des documents d'urbanisme.

Cordialement.

Reviewer 4 Report

Figure need  higher resolution, such as fig.5

Minor editing of English language required

Author Response

Dear Reviewer

Thank you for taking the time out of your busy schedule to provide us with quality comments on our manuscript. Taking your comments into account, please find the answers in the lines below.

Regarding the resolution of the figure you mentioned, we'd like to inform you that it doesn't depend on us, but on the software and especially the packages used. Otherwise, we have shot all the figures at high resolutions. Thank you

As far as the minor modification of the English language is concerned, we're taking care of that. Thank you

Round 2

Reviewer 1 Report

Specific editing suggestions:

Section 1.2

Line 74:  start a new paragraph where it says, “Increased heat…..”

Line 92:  start a new paragraph where it says, "On the other hand…..”

Section 1.3

Line 110:  better to say “economic development sectors” rather than several development sectors.

Line 118:  should say “was the focus of another study” or something similar.  Right now there is only a reference number.

Line 154:  suggest “evidence-based” scientific knowledge, rather than “notorious” scientific knowledge.  This is not a proper use of the word “notorious.”

Line 170:  suggest writing the following  “(sustainable cities and communities or Make cities inclusive, safe, sustainable, and resilient.)” at the end of this line

Line 171: add a reference at the end of this line  (Agenda 2063: The Africa We Want. | African Union (au.int) or explain that it is Agenda 2063 of the African Union.

Section 2.2

Line 193:  It might be better to refer to the category of interviewees as “leadership” rather than “elites” because elites has a negative connotation and typically refers to a group that is highly restricted and difficult to join.  Authors may not want to inflict that assumption on local leaders.

Lines 207 and 208:   The Exposure Index and the Impact Index are defined in exactly the same way.  Shouldn’t they be different?      

Section 3.2

Paragraph 3:  this is very confusing and should be rewritten for the sake of clarity. The paragraph starts by saying that this group is very different than the 1st group.  They are described as “less” familiar with climate risks.  But their perceptions seem to be the same.  It says that flood and temperature are perceived as “average” risks rather than “high risks.”  Is this the difference: average vs high?  If so, that sentence should be re-written to make this more clear.  They recognize the same 2 risks as most prominent.  So why is this important?  Are you saying that the only difference in their perception is about high winds?  If so, perhaps you could specific write “why do they have a different perception regarding the risk of high winds? “ 

Table 4

Isn’t the actual story that the perceptions of the younger group versus the older group are that all risks are less of a problem?  Perhaps age is the key difference?

Table 5

Please make clear that all of the impacts described in the Table come from the interviews of Urban Leaders.  If this isn’t the case, please make clear at the end of the Table with an asterisk * in the title where they do come from.

Section 3.5

This description is unclear.  From where did you derive the principal components?  What are they?

What does it mean to say that they are “negatively and positively correlated?” Are you saying that some adaptation measures are viewed as positive (helpful) measures and others are viewed as negative (not helpful measures)?

Table 6

I do not really understand Table 6.  But perhaps it is just my limitation in understanding this methodology.   The sentence underneath Table 6 is in French and should not be there.

Section 4 Discussion

Fix this “These results are in line with the work of [50]”.  Who is 50?

4.1  This section is too long and much is not terribly relevant or interesting.

               First bullet: I believe you could delete everything here after the first 3 sentences

               Second bullet:  I believe you can delete all here after the first 2 sentences.  You could leave in the sentence that starts “According to Zhou and Williams……”

4.2   This section is more interesting and relevant.

4.3  Good section

5     Leave as is. 

Author Response

Dear Reviewer

We would like to thank you for all the recommendations you keep making to ensure the perfection of the manuscript. Please find the answers to your remarks in the lines below.

Section 1.2

Line 74 : start a new paragraph where it says, “Increased heat…..”

Response : Done

Line 92 : start a new paragraph where it says, "On the other hand…..”

Response : Done

Section 1.3

Line 110 : better to say “economic development sectors” rather than several development sectors.

Response : Done

Line 118 : should say “was the focus of another study” or something similar. Right now there is only a reference number

Response : Done

Line 154 : suggest “evidence-based” scientific knowledge, rather than “notorious” scientific knowledge. This is not a proper use of the word “notorious.”

Response : Done

Line 170 : suggest writing the following “ (sustainable cities and communities or Make cities inclusive, safe, sustainable, and resilient.)” at the end of this line

Response : Done

Line 171 : add a reference at the end of this line (Agenda 2063 : The Africa We Want. African Union (au.int) or explain that it is Agenda 2063 of the African Union.

Response : Done. We have added a reference

Section 2.2

Line 193 : It might be better to refer to the category of interviewees as “leadership” rather than “elites” because elites has a negative connotation and typically refers to a group that is highly restricted and difficult to join. Authors may not want to inflict that assumption on local leaders.

Response : The Exposure Index (IE) is the sum of the impacts of climate risks on the urban issue (read the sum horizontally). Impact Index (II) is the sum of the impacts of a climate risk on urban issues (read the sum vertically). (see table 2).

Section 3.2

Paragraph 3 : this is very confusing and should be rewritten for the sake of clarity. The paragraph starts by saying that this group is very different than the 1st group. They are described as “less” familiar with climate risks. But their perceptions seem to be the same. It says that flood and temperature are perceived as “average” risks rather than “high risks.” Is this the difference : average vs high? If so, that sentence should be re-written to make this more clear. They recognize the same 2 risks as most prominent. So why is this important ? Are you saying that the only difference in their perception is about high winds ? If so, perhaps you could specific write “why do they have a different perception regarding the risk of high winds? “

Response : The second group of respondents differs from the first in terms of the homogeneity of responses and, above all, socio-demographic parameters. For the first respondents, we have the highest level of education, the elderly and public and private sector employees. For the second respondents, we have the level of education (secondary and primary), young people, adults and professional craftsmen. The youth and level of education of the second respondents mean that they have less advanced knowledge of climate risks than the first. Both groups perceive climate risks, but to different degrees. We have removed the word "different" from line 290.

Table 4 : Isn’t the actual story that the perceptions of the younger group versus the older group are that all risks are less of a problem ? Perhaps age is the key difference ?

Response : All the risks perceived by both groups are problematic. However, the level or degree of perception differs from the youngest to the oldest.

Table 5 : Please make clear that all of the impacts described in the Table come from the interviews of Urban Leaders. If this isn’t the case, please make clear at the end of the Table with an asterisk * in the title where they do come from.

Response : Done. We mentioned this in the table title

Section 3.5

This description is unclear. From where did you derive the principal components? What are they?

What does it mean to say that they are “negatively and positively correlated ?” Are you saying that some adaptation measures are viewed as positive (helpful) measures and others are viewed as negative (not helpful measures) ?

Table 6

 I do not really understand Table 6. But perhaps it is just my limitation in understanding this methodology. The sentence underneath Table 6 is in French and should not be there.

Response : The adaptation measures cited are all useful. Positive and negative correlations are derived from the statistics and packages of the R software used. This in no way affects the coping strategies cited by the population.  Thus, the principal component analysis is also derived from the specific packages used to process the qualitative data obtained. The data in Table 6 are the same as those projected in Figure 5, and are presented in the table for the reader's convenience. We have corrected the sentence in French.

Section 4 Discussion

Fix this “These results are in line with the work of [50]”. Who is 50 ?

Response : Done. We have corrected

4.1 This section is too long and much is not terribly relevant or interesting. First bullet : I believe you could delete everything here after the first 3 sentences Second bullet : I believe you can delete all here after the first 2 sentences. You could leave in the sentence that starts “According to Zhou and Williams……”

Response : Done. We have removed your request
